# Oil Heat Treatment of Wood—A Comprehensive Analysis of Physical, Chemical, and Mechanical Modifications

**DOI:** 10.3390/ma17102394

**Published:** 2024-05-16

**Authors:** Eleni Mandraveli, Andromachi Mitani, Paschalina Terzopoulou, Dimitrios Koutsianitis

**Affiliations:** 1Department of Forestry, Wood Science and Design, University of Thessaly, GR-43100 Karditsa, Greece; emandraveli@uth.gr (E.M.); dkoutsianitis@uth.gr (D.K.); 2Department of Harvesting and Technology of Forest Products, Faculty of Forestry and Natural Environment, Aristotle University of Thessaloniki, GR-54124 Thessaloniki, Greece; terzopouz@for.auth.gr

**Keywords:** wood heat treatment, oil treatment, physical modifications, chemical modifications, mechanical properties

## Abstract

Wood, a natural material with versatile industrial applications, faces limitations such as low dimensional stability and decay resistance. To address these issues, there has been significant progress in wood modification research. Oil heat treatment has emerged as an effective method among environmentally friendly wood treatment options. Studies have indicated that treating wood with hot vegetable oils yields superior properties compared to traditional methods involving gaseous atmospheres, which is attributed to the synergistic effect of oils and heat. This comprehensive review investigates the physical, chemical, and mechanical modifications induced by the oil heat treatment of wood, along with its impact on biological durability against biotic agents. The review synthesizes recent research findings, elucidates underlying mechanisms, and discusses the implications for wood material science and engineering.

## 1. Introduction

### Wood: General Information

Wood is a natural material, excellent for several uses in the industrial sector such as construction, furniture, packaging, chemical industry [1], the production of wood-based products [2], and as an engineering material used in buildings and bridges [3]. It is considered of high economic importance and it can be simply processed [2,4], it is non-toxic, inexpensive, and accessible [5].

However, wood is a natural renewable product that comes from different trees, and its use may impose several limitations such as dimensional instability, low resistance to biotic and abiotic damages [6], and differentiations in its mechanical properties [5]. These limitations play a critical role in its financial significance as a raw material and several modifications are applied to increase its functionality [7]. The main approach to improve wood properties is to modify its cell wall constituents, e.g., cellulose, hemicellulose, lignin, and extractives [2]. In the past, some of the wood properties were partially ameliorated with traditional methods like chromated copper arsenate (CCA) and creosote, but their use nowadays is strictly regulated in the EU because they are toxic, dangerous for the environment, high-cost registered, and not well accepted by the public [7,8].

Over the last decades, there has been a serious interest throughout the academic, industrial, and social sectors in wood modification. This can be seen in changes in silvicultural practices, such as the adoption of sustainable forestry techniques to mitigate the impacts of climate change on forest ecosystems [9]. Additionally, there has been an expansion of knowledge regarding the use of rare species, driven in part by concerns about biodiversity loss exacerbated by changing climatic conditions. Increased environmental awareness of the chemicals used in the wood product industry reflects a growing recognition of the role of greenhouse gas emissions and pollutants in contributing to climate change. Moreover, there is a notable interest in adding value to sawn timber, spurred by the need for climate-resilient infrastructures and sustainable building materials in the face of shifting environmental conditions. Compliance with EU regulations and addressing the international issue of climate change further underscore the interconnectedness of the environmental, economic, and social factors shaping the wood product industry [10,11].

European countries are committed to making the transition to climate neutrality by 2050, and this is going to happen by investing in technical solutions and transferring from carbon-intensive raw materials to biobased resources. The timber industry is now, more than ever, investing in methods to improve wood properties in order to be more appealing for consumers and take a share of the global market [12].

The aim of this paper is to conduct a review of the physical, chemical, and mechanical property alterations of wood heat-treated in oil. Moreover, it aims to review the procedures followed in wood thermal modification using several types of oils as a heat-transferring medium and to examine the results of these methods and systematically compare them.

## 2. Wood Modification

### 2.1. Definition

According to Hill (2006), wood modification is defined as the procedure that “involves the action of a chemical, biological or physical agent upon the material, resulting in a desired property enhancement during the service life of the modified wood” [13].

### 2.2. Processes

Wood modification is the process of changing the physical or chemical properties of wood, optimizing its performance for exterior applications, for the dimensional stability of wood for interior applications [14,15], or for ameliorating the acoustical properties of wood for musical instruments [16]. The modification of wood has undergone significant evolution over time, resulting in the development of innovative solutions that enhance the performance and durability of wood products.

Wood modification is classed according to the procedure used to modify the properties of the wood. Table 1 provides a clear overview of the three primary classifications of wood modification techniques, which can be further divided into chemical, thermo-hydromechanical, and physical categories. These techniques are widely used in the wood industry and have proven to be highly effective. However, in addition to these well-established modification approaches, several emerging modifications have been developed with the potential to revolutionize the wood industry. Nevertheless, these novel techniques remain largely unexplored and are currently being employed mainly in other fields such as agriculture, energy conversion, and food processing.

Chemical modification can be described as cell wall modification or the filling of large cell cavities. What is aimed to be achieved is the moisture content decrease, the expansion of cell walls, and changes in the molecular structure of cell walls [13]. Table 2 presents the wood modification methods which have achieved commercial production.

Significant development is currently being made in the wood modification industry. There are several commercial techniques of wood modification such as acetylation, furfurylation, thermal modification (heat treatment), and resin impregnation/polymerization [12].

This section provides a detailed overview of three prominent wood modification techniques: acetylation, furfurylation, and resin impregnation/polymerization. Each process will be examined in terms of its historical development, contemporary relevance, and the impact it has on altering wood properties [24].

Acetylation is a chemical modification process that involves the introduction of acetyl groups into the wood’s cell wall structure. This technique dates back to the early 20th century when researchers first explored the potential of acetylation to improve wood’s dimensional stability and resistance to biological degradation. Currently, acetylation is recognized for its ability to significantly enhance wood properties, including reduced swelling and shrinkage, increased resistance to decay, and improved durability in outdoor environments. By effectively blocking sites within the wood where moisture can be absorbed, acetylation creates a more stable material that is less prone to warping, cracking, and fungal attack. However, challenges such as high production costs and the need for specialized equipment and expertise limit its widespread adoption in the industry [25]. Indeed, alternative methods for acetylating wood have been explored to expand the range of options and overcome certain limitations associated with traditional acetylation processes. Some of these alternative methods include ketene or diketene acetylation [26]. Researchers have investigated the use of ketene or diketene as acetylating agents for wood modification [13]. Ketene is a highly reactive compound that can react with wood cellulose to introduce acetyl groups. These methods offer potential advantages such as improved reaction efficiency and control over acetylation levels. Acetylating wood with acetic acid under harsh conditions, such as elevated temperatures and pressures, has been explored as an alternative method. These conditions promote more rapid and extensive acetylation reactions, leading to the enhanced modification of wood properties. However, the use of harsh conditions may also pose challenges in terms of process control and energy consumption. Another alternative method involves the direct reaction of wood with acetyl chloride to introduce acetyl groups. While this method can yield efficient acetylation, it is often associated with the release of hydrochloric acid as a by-product. The generation of hydrochloric acid may pose environmental and safety concerns, requiring appropriate measures for handling and disposal.

Furfurylation: This technique emerged in the mid-20th century and has since undergone significant development to improve its effectiveness and applicability. Furfurylation offers several benefits, including enhanced dimensional stability, increased resistance to decay, and a darkened appearance that resembles traditional hardwoods. It is particularly well suited for outdoor applications where durability is paramount, such as in decking, cladding, and marine environments. Nevertheless, concerns persist regarding the environmental impact of furfuryl alcohol and the cost-effectiveness of large-scale production. Ongoing research aims to address these challenges and further optimize the furfurylation process for broader industry use [27,28]

The process of resin impregnation involves saturating wood with synthetic resins, which are then cured to form a hardened polymer within the wood structure. This process has a long history, dating back to the early 20th century, and has undergone continuous refinement to meet evolving industry needs. Resin impregnation offers numerous advantages, including increased strength, improved dimensional stability, and enhanced resistance to moisture, insects, and decay. It is a common practice in the production of engineered wood products such as plywood, laminated veneer lumber, and glue-laminated timber. However, concerns remain regarding the formaldehyde emissions, environmental sustainability, and cost-effectiveness of this process. Research is underway to develop alternative resin systems and processing methods that address these issues while maintaining the performance benefits of resin impregnation [29,30,31].

On the other hand, thermal modification is the exposure to heat that results in higher dimensional stability and increased strength towards biotic enemies [32]. However, the cell wall changes that occur during that process induce serious loss in the mechanical properties [33] and this limits the application of modified wood as an engineering material [4].

### 2.3. Thermal Modification

Thermal modification is one of the oldest methods used to modify wood properties. A high-temperature drying procedure was studied first in 1920 and showed significant results in moisture content decrease, and thus, in wood dimensional stability increase [34]. The use of heat in wood processing has played an important role in ensuring its life-long service. Nowadays, apart from thermal modification, there are several different wood-heating processes such as wood drying; heating in the absence of air like pyrolysis and thermolysis; heating in the presence of air like combustion; complete combustion with full access to oxygen; and incomplete combustion with limited oxygen access [21]

Thermal modification processes are used to alter the whole solid wood volume, diffusing the heat equally. It is faster and more manageable to do so, without chemical additions, in an oxygen-free environment at temperatures between 160–240 °C [35]. To avoid oxygen access, a shield media such as steam, nitrogen gas, vegetable oil, or vacuum is used [14]. The cell wall changes during heat exposure, inducing a loss in strength and ductility that may cause problems in load-bearing applications [36] and color darkening ([33,34], but the wood exhibits serious dimensional stability increase (Biziks et al., 2015) [37] and biological strength [6]. Thus, timber with low natural durability can be heat-treated to increase its decay resistance to be suitable for non-structural and outdoor applications and to augment its economic value [38,39].

At the end of the life cycle of the heat-treated wood, it can be recycled without severe environmental impact, to the contrary of chemically treated wood impregnated with biocidal active ingredients [34]. That is the reason why the thermal modification of wood has been recognized as environmentally friendly and the most commercially successful method that improves wood properties [40].

The property alteration extent after the thermal modification depends on the wood species, the method used, the type of environment, and the process schedule [8]. Nowadays, there are various commercial treatment processes available, differing in inert atmosphere and curing conditions [39].

As shown in Table 3, the most commercialized modification processes as well as the conditions applied started appearing in Europe around 1970. The first process was Feuchte-WärmeDruck (FWD), in Germany ca. 1980; however, it was not greatly industrialized. In the same period, in The Netherlands there was the PLATO process and in France the Bois Perdure, and they were very similar to the rectification process. In the 1990s, ThermoWood^®^ was established in Finland, and by 2016 it had grown all over Europe and around the world. Another process was TERMOVUOTO, a thermo-vacuum modification that showed better results on the mechanical properties of wood [34]. One of the most recent processes is OHT (oil heat treatment), which was developed in Germany around 2000 [25].

In 2017, ThermoWood Association reported that the global production volume was 193,700 m^3^ and Thermory^®^ (Tallinn, Estonia) in Estonia reported an additional 160,000 m^3^ of thermally modified timber. The estimated production of heat-treated wood in Europe is 695,000 m^3^ [5]). The process seems to be very popular, mainly due to the fact that it can be commercialized on small scales [12]

### 2.4. Oil Heat Treatment

Oil is used in heat treatment as an alternative way of treating wood without the use of chemical preservatives, and thermal modification in oil baths is an effective environmentally friendly wood treatment method [45,46].

The requirement of oxygen absence in heat treatment led to the use of various vegetable oils as a heat-transfer and oxygen-excluding medium. The boiling point of many of the vegetable oils used is usually higher than 260 °C, so the use of oils as heating media is relevant [8].

It is proven that wood properties when using hot vegetable oils are better compared to when heated in a gaseous atmosphere, and that is a result of the synergy effect of oils and heat [8,47,48]. The hydrophobic properties and resistance to fungi of thermal-oil-modified wood were achieved by the heat treatment and also from the thin-coated layer formed by the vegetable oil. Oils can decrease water uptake and thus improve the water repellence of wood. To achieve that, it is required to highly retain and deeply impregnate oil into the wood structure to lengthen the duration of life performance in external conditions [49].

There are various oil heat treatment methods that offer distinct approaches to enhancing wood properties for different applications, like OHT originating from Menz Holz in Germany. This process involves subjecting the wood to temperatures of 180 °C–260 °C using oils like rapeseed, linseed, or sunflower oil as a heat transfer medium. The process occurs in a closed vessel where the hot oil circulates around the wood. This method effectively separates oxygen from the wood, improving its durability and stability. The desired temperature is maintained in the core of the wood for 2–4 h [48]. Moreover, the Royal process (combined impregnation) developed by Hager combines impregnation with a wood preservative followed by treatment with hot oil. It offers both fungicidal effects from the preservative and hydrophobic properties from the oil [50]. Two vessels are typically used for the two-step process: the Lowry process for impregnation and oil impregnation for hydrophobizing the wood surface to inhibit preservative leaching. An other method is the Bi-oleothermal Process which was developed by CIRAD in France; this process involves two phases: a hot bath and a cold bath, both occurring at atmospheric pressure. The wood specimens are first dipped in a hot oil bath and then in a lower-temperature oil bath. The hot bath creates overpressure inside the wood, while the cold bath causes water condensation, creating a vacuum inside the wood that facilitates oil impregnation into the deepest layers [51]. Other methods exist, such as thermally modifying wood before oil impregnation, in-treatment cooling where samples are cooled in the oil bath, and post-treatment cooling where samples are cooled at various intervals after heat treatment in the oil bath.

The oil treatments of wood typically involve the application of a specific quantity of oil onto the wood surface, allowing it to penetrate the material. The amount of oil absorbed depends on various factors such as the type of wood, its porosity, and the specific oil used. Generally, the increase in mass due to oil absorption can range from a few percentage points to as much as 10% or more of the original wood weight, depending on the treatment method and the desired outcome [52].

After treatment, there are typically procedures to remove excess oil from the wood surface. This can involve wiping off any surplus oil or using methods such as sanding to smooth out the surface and remove excess oil buildup. In some cases, if the wood is heavily saturated with oil, it may be necessary to remove the outer layer of the wood to expose a fresh surface that is not overly oily. However, this depends on the specific requirements of the application and the desired level of oil saturation in the wood [10].

There has been significant research conducted on the subject and there are several oil heat treatment procedures available in the literature. These procedures differ in wood species, sample dimensions, oil type, heat temperature, and process time [53].

However, most research focuses on examining the results of the process on the physical (EMC and hygroscopicity), chemical, and mechanical (strength or brittleness) properties of wood that are mainly influenced by oil heat treatment [54].

### 2.5. Changes in Chemical Composition

As a result of heat treatment, the wood undergoes significant changes in its chemical composition; firstly, by degrading its amorphous carbohydrates, acetic acid is formed [55], cellulose crystallinity [56] and lignin content are increased [57], and there is a decrease in the extractive content due to the evaporation of the volatile compounds [58].

Most of the literature focuses on the effects of chemical changes on the physical properties of wood. However, Wahab et al. (2012) took samples of a planted 15-year-old *Acacia* hybrid cut into dimensions 300 mm × 100 mm × 25 mm (length × width × thickness) and heat-treated them in palm oil at 180 °C, 200 °C, and 220 °C for 30, 60, and 90 min, respectively. Afterwards, the samples were milled into sawdust and analyzed resulting in a significant decrease in cellulose and holocellulose and an increase in hemicellulose and lignin content. The same experiment was conducted some years later [59,60] on planted *Acacia mangium*, common name Silver Wattle, showing a decrease in holocellulose and α-cellulose as well as hemicellulose and extractive contents as the heat treatment lasted longer.

In their study, Umar et al. (2016) subjected rubberwood (*Hevea brasiliensis*) samples measuring 300 mm × 20 mm × 20 mm (in longitudinal, radial, and tangential dimensions) to heat treatment in palm oil. The temperature ranged from 172 °C to 228 °C, with corresponding treatment durations of 95 to 265 min, respectively. As the temperature and duration of the treatment increased, there was a decrease in cellulose and hemicellulose content, coupled with an increase in lignin content [61].

Lee et al. (2018) experimented on particleboards (PB) made of rubberwood particles that were bonded with urea formaldehyde resin. The PB samples were put in palm oil for 24 h at 25 °C and then heat-treated at 180 °C, 200 °C, and 220 °C for 2 h. The research showed the same results as the one mentioned above, a degradation of hemicellulose and an increase in lignin content [62].

Okon and Udoakpan (2019) used silicone oil to heat-treat Masson pine (*Pinus massoniana*) at 150 °C, 180 °C, and 210 °C for 2 h, 4 h, and 8 h, resulting in a significant decrease in polysaccharides and ash contents and a significant increase in lignins and extractives as the treatment time increased. Also, the study shows an increase in the crystallinity index and the contact angle of the wood [45].

Xing, Li, and Wang (2020) cross-examined three different heat treatment processes on Gmelin larch (*Larix gmelinii Rupr.*). In the oil heat treatment process, the specimens were cut in 10 mm × 9 mm × 9 mm (longitudinal, radial, and tangential dimensions) and heated in canola oil, resulting in increased degradation as the temperature increased. There was an elevation of the mass due to oil uptake and the appearance of triglyceride fatty acids, deacetylation, depolymerization, cross-linking, and recondensation [63].

Hao et al. (2021) examined moso bamboo samples of 300 mm × 30 mm × 8 mm (longitudinal × tangential × radial) in methyl silicon oil at 140 °C, 160 °C, 180 °C, and 200 °C for 2 h, 4 h, and 6 h, respectively, and showed that with increasing heat treatment temperature and time, there is a gradual decrease in the content of cellulose and hemicellulose while there is an increase in the relative content of lignin which is more thermally stable [64].

Mastouri et al., 2021, in their study investigated the effects of heat treatment using rapeseed oil and silicone oil on wood samples. The wood samples were taken from the sapwood of 30-year-old *P. deltoides* trees, the common name of which is the eastern cottonwood or necklace poplar, with the specific dimensions of 30 × 30 × 20 mm^3^ (longitudinal × radial × tangential) and immersed in oil baths at room temperature and treated in a high-temperature oven for 4 h. When the temperature reached the 190 °C, the Fourier-transform infrared spectroscopy (FTIR) analysis revealed significant modifications in the chemical composition of the wood after treatment. Changes such as dehydration, hydrolysis, oxidation, decarboxylation, and transglycosylation were observed. The heat treatment in the oil bath led to the degradation of cellulose and lignin, as indicated by the shifts in the prominent peaks associated with these components. The ratio of FTIR signals related to changes in wood polymers is correlated with the contact angle of water droplets. An increase in the ratio of certain peaks indicates higher crystalline cellulose content and lignin content, which corresponds to lower surface wettability [65].

Bessala et al. (2022) conducted a study to assess the effects of heat treatment on *Afrormosia* and *Newtonia* wood under both air and palm oil conditions. The wood samples were subjected to heat treatment at three different temperatures (160 °C, 180 °C, and 200 °C) for two hours. The study evaluated various properties such as color changes, dimensional stability, hygroscopicity, and chemical structure for each treatment condition. The samples treated in palm oil appeared darker compared to those treated in air. FTIR analysis revealed a decrease in hydroxyl groups with increasing wood treatment temperature, indicating alterations in the chemical structure of the wood. The heat treatment at high temperatures led to the irreversible degradation of hemicelluloses and cellulose, primarily through the cleavage of the chemical bonds within these polymers, including β-O-4 linkages in lignin and the breaking down of the methoxyl groups from lignin. Furthermore, heat treatment with palm oil introduced modifications in carbohydrates, particularly in hemicelluloses, through oxidation processes. Palm oil treatment formed an elastic film on the wood surface, enhancing wood hygroscopicity by reducing water penetration into the wood structure. This film formation also contributed to the darker appearance of the palm oil-treated wood compared to the air-treated wood, resulting from the interaction between the unsaturated fatty acids in palm oil and the wood surface [66].

He et al. (2023) investigated the effects of heat treatment and tung oil impregnation on the dimensional stability, surface wettability, microstructure, pore structure, and chemical composition of Siberian elm (*Ulmus pumila* L.) wood. They used the heartwood of elmwood with an oven-dried density of 0.63 ± 0.03 g/cm^3^ with the dimensions of 20 × 20 × 20 mm^3^ (longitudinal × radial × tangential) for moisture absorption and water absorption tests. The samples were randomly divided into 25 groups, each containing 15 samples. The experimental design consisted of the control group, air-heat-treated wood (AHT): Twelve groups underwent heat treatment in hot air and Tung oil-heat-treated Wood (OHT): The remaining twelve groups were modified using tung oil. The heat treatment parameters were varied as temperature: 120 °C, 150 °C, 180 °C, and 210 °C for 2 h, 4 h, and 6 h. The temperature and duration combinations were applied systematically to investigate their effects on the wood samples. The FT-IR analysis revealed significant chemical changes in the wood samples after heat treatment and tung oil impregnation, including reductions in hydroxyl groups, the degradation of lignin and carbohydrates, and the presence of tung oil, all contributing to improved dimensional stability and reduced water absorption properties [67].

## 3. Physical Property Alterations

### 3.1. Equilibrium Moisture Content (EMC) and Dimensional Stability

Chemical changes result in several alterations of the physical properties of wood. The decrease in the amount of hydroxyl groups (Jamsa and Viitaniemi, 2001) and the increase in cellulose crystallinity lead to the reduction in EMC and to better dimensional stability [68,69].

Another study of *Bambusa vulgaris* var. *Striata* was conducted by Izran et al. (2012), where samples of 20 cm long × 5 cm wide were heated in crude palm oil at 160 °C, 180 °C, and 200 °C for 10 min. This treatment resulted in the increase in the dimensional stability of the bamboo tree by removing moisture and minimizing hygroscopicity [70].

Bak and Nemeth (2012) heat-treated poplar wood samples of 220 × 18 × 40 mm (longitudinal, radial, and tangential dimensions) in sunflower, linseed, and rapeseed oils at 160 °C and 200 °C for 2 h, 4 h, and 6 h and came to the conclusion that EMC decreased and dimensional stability increased [71].

Similar conclusions appeared in Bazyar’s study (2012) where he took samples from aspen trees and cut them into 50 cm × 20 cm × 20 cm and 20 cm × 20 cm × 20 cm. The specimens were immersed in hot linseed oil for 4.5 h and 6 h at 190 °C, 205 °C, and 220 °C, respectively. The results of this study showed that the EMC of heat-treated samples was significantly less when the temperature was higher than 190 °C, but with no relevance to the treatment time. The shrinkage of heat-treated samples was significantly lower when the temperature was higher [72].

Tomak et al.’s (2011) research shows that changes in dimensional stability depend on the species studied. Both beech wood (*Fagus orientalis* Lipsky) and Scots pine (*Pinus sylvestris* Fastigiata) samples were treated at 160 °C for 30 min and beech has higher water absorption than Scots pine because it retains more oil [73].

Bal (2016) tested beech (*Fagus orientalis* L.) sapwood specimens of 20 mm × 20 mm × 30 mm in hot air and in hot sunflower oil at atmospheric pressure at the temperatures of 160 °C, 190 °C, and 220 °C for 2 h, showing that EMC in oil-heat-treated groups was lower than in the air heated ones. The lowest EMC was at 220 °C, a similar result to the research mentioned above (Bazyar, 2012; Bak and Nemeth, 2012), and it appears that the higher the process temperature, the lowest the EMC, and that is a result of the free hydroxyl group degradation (Almeidae et al., 2009) [74]. It is safe to conclude that oil as a heat medium affects significantly the results of the thermal treatment [75].

Dimensional stability was also improved in Wang and Cooper’s (2005) studies, where palm oil, soy oil, and slack wax are used on white spruce wood of 10 mm × 25 mm × 25 mm (longitudinal, radial, and tangential dimensions) at 200 °C and 220 °C for 2 h and 4 h, respectively, resulting in lower EMC (especially in the slack wax bath) and improved dimensional stability. There was no significant difference in water absorption among the three treatment temperatures; however, at 220 °C there was a greater improvement in dimensional stability than at 160 °C or 100 °C. These improvements are mainly due to chemical changes that occur in high temperatures, although the oil bath plays a significant role in benefiting wood, especially by decreasing water absorption [53].

Dubey et al. (2011) studied the sapwood of Monterey pine (*Pinus radiata*) samples of 200 mm × 90 mm × 35 mm (longitudinal, radial, and tangential dimensions) in a raw linseed oil bath at 180 °C for 3 h and then in a used linseed oil bath at 180 °C for 3 h, 9 h, 15 h, 21 h, and 27 h, respectively. Dimensional stability was improved; however, this was mainly due to the thermal treatment and not the oil bath [10].

Rotary-peeled aspen (*Populus tremuloides*) wood veneers of 700 mm × 700 mm × 3.2 mm were heat-treated at 200 °C and 220 °C and cut again in 50 × 50 mm and treated in hot canola oil at 180, 200 °C, and 220 °C for 1, 2, and 3 h (Fang et al., 2011). This study showed that dimensional stability was improved after the oil heat treatment and compression set recovery reduced as the temperature and duration of the oil heat treatment increased.

Mukam et al. (2012) took samples of ~80 × 20 × 50 mm of Ayous wood (*Triplochiton scleroxylon* K. Schum) and Sapelli (*Entandrophragma cylindricum*), and heat-treated them in two palm oil baths at 200 °C and 23 °C, respectively, for 30 min each [76].

EMC, swelling, and shrinkage were reduced after the treatment, though this may be a result of the heat treatment and not of the oil bath.

Fir wood (*Abies* sp.) samples of 300 mm × 50 mm × 50 mm were heat-treated in soybean oil and maleic anhydride (MA) at 100 °C, 120 °C, 140 °C, 160 °C, and 180 °C for three 30, 60, and 180 min. The MA-heat-treated specimens at 160 °C for 60 min had the best results in their physical properties in comparison with soybean oil heat treatment [77].

Lee et al. (2018), who were mentioned in the previous chapter, concluded that thickness swelling on particleboards improved only after oil heat treatment at 220 °C [62].

European beech (*Fagus sylvatica* L.) oil-heat-treated wood was studied by Baar et al. (2021) to obtain info concerning physical and mechanical properties as well as decay resistance. Specimens of 20 mm × 20 mm × 300 mm were cut with the growth rings parallel to the edges and were impregnated in hemp oil for 2 h, heat-treated at 200 °C for 3 h, and heat-treated in hemp oil bath at 200 °C. An improvement in dimensional stability of 10%, 20%, and 25%, respectively, was a result of swelling reduction [8].

Düzkale Sözbir et al. (2021) took Stone pine (*Pinus pinea* L.) samples and after thyme oil bath for 5 min and 3 weeks resting in air conditions, the samples were heat-treated at 150 °C for 1 h. The results showed improvements in the physical properties and especially in water absorption. Var et al. (2021) took Turkish pine (*Pinus brutia* L.) sapwood samples of 15 mm × 30 mm × 30 mm and heated them in a hot oil bath (castor oil, flaxseed oil, and mixed oil) at 110 °C for 6 h and cold oil bath for 2 h at 23 °C. This resulted in better dimensional stability and water repellence [78,79].

Mastouri et al., 2021, investigated the impact of two different oil heat treatment methods, namely SOHT (silicone oil heat treatment) and ROHT (rapeseed oil heat treatment), on various properties of (*P. deltoides*) wood with dimensions of 30 mm × 30 mm × 20 mm. Both treatments significantly affected the water absorption and dimensional stability of the wood samples. Specifically, the SOHT-treated samples showed lower water absorption and volume swelling compared to the ROHT-treated samples, especially when treated in silicone oil. To be more exact, the ROHT samples showed a 10% decrease in volume swelling from 3 h to 24% and a 2% decrease from 24 h to 96 h. On the other hand, the ROHT-treated samples showed a 5% decrease in volume swelling from 24 h to 96 h. The water absorption decreased by 36.36% in the SOHT-treated samples from 3 h to 24 h and by 71% from 24 h to 96 h and for the ROHT-treated samples decreased by 57.14% from 3 h to 24 h and by 67% from 24 h to 96 h [65].

This reduction in water absorption is attributed to the penetration of oil into wood cavities, leading to decreased hygroscopicity. Contact angle measurements showed that the SOHT-treated samples had higher water repellency compared to the ROHT-treated samples. This is attributed to the intensive degradation of hemicelluloses in silicone oil medium and the higher water repellency of silicone oil compared to rapeseed oil. The study suggests that the improved properties of wood treated with SOHT are less affected by soaking time and leaching procedures, ensuring the long-lasting protection of wood compared to ROHT treatment [65].

The results of the study by Bessala et al. (2022) which used air and palm oil heat treatment showed that heat treatment in air improved dimensional stability, as indicated by reduced tangential and radial swelling coefficients at 200 °C for both Afrormosia and Newtonia wood. Loss in mass was observed only for the thermal treatment under air conditions, with maximum losses occurring at 200 °C. The heat treatments under air at 200 °C also reduced water absorption for both the wood species. The treatment in palm oil further improved stability and hygroscopicity compared to the air treatment, with significant reductions in tangential and radial swelling coefficients observed at 200 °C [80].

The research of Kaya et al. (2023) used heat treatment with linseed oil on the wood of Mediterranean cypress and field maple. After treatment at 240 °C, water absorption from the Mediterranean cypress wood at the end of day 30 (720 h) was 36% [81]. In contrast, the reference group (non-treated wood) had a water absorption of 80% in the same time frame. This indicates a significant improvement in water absorption, with a 55% reduction compared to the untreated wood. The control group (untreated wood) still exhibited water absorption, albeit at a higher rate. For Mediterranean cypress, the control group had a water absorption of 65%, indicating a 35% improvement due to heat treatment. Similarly, for field maple wood, the control group had a water absorption of 72%, showing a 28% improvement due to heat treatment. The results suggest that impregnation with linseed oil (LO) significantly contributed to reducing water absorption in heat-treated wood and enhanced the hydrophobic properties of wood. The findings align with similar studies conducted by [43,82,83], which also demonstrated the effectiveness of impregnation treatments in reducing water absorption in wood.

### 3.2. Color Changing

Color, as well as gloss and texture, is very important for the appearance and thus the commercial value of wood [8]. Color darkening seems to be a property change that can be either desirable or undesirable [12]; however, the resemblance of dark wood to expensive wood attracts more consumers [84]. Wood color is mostly influenced by the temperature and duration of the thermal process [85].

The method used for the assessment of color changing on wood after specific modifications is the CIE L× a× b system, where the values obtained from the wood surface are represented in three axes, L axis for lightness (0 is black and 100 is white), a axis for green–red (−a is green and +a is red) and the b axis for blue–yellow (−b is blue and +b is yellow) [86]. The final color difference is determined by the following equation:$$\Delta Ε={{[\Delta L}^{2}+{\Delta A}^{2}+{\Delta b}^{2}]}^{1/2}$$

**The three-dimensional CIE L× a× b× color space** [85]

Sailer et al. (2000) thermally treated spruce (*Picea abies* L. Karst.) and Scots pine (*Pinus sylvestris* L.) in linseed or rapeseed oil bath at 180 °C and 220 °C. The sample surfaces were colored uniformly due to the even distribution of wood resins in the oil bath that surrounded them [47].

Mukam Fotsing and Simon Fokoua (2012) studied Sapelli and Ayous in a palm oil bath at 200 °C for 30 min and then at 23 °C, showing significant color changes. Tjeerdsma et al. (2005) studied Norway spruce and Scots pine specimens with oil heat treatment at 180 and/or 200 ° for 3 h. The color was affected and the changes depended on the oil type used in the process [76].

Bak and Nemeth (2012), mentioned above, showed that the wood samples had a significant darkening in color and the extent of the changes was affected by the species studied [71,87].

The wood samples were also darkened in Wahab et al. (2017) as well as in Baar et al.’s (2021) research [59,60], while Lee et al. (2018) concluded that the higher the treatment temperature, the darker the color [86].

The findings reported by Kim et al. (2018) provide insights into the effects of air heat treatment on *Paulownia tomentosa* wood at different temperatures. As the temperature of air heat treatment increased (ranging from 160 °C to 220 °C), the lightness (L) of the *Paulownia tomentosa* wood decreased. This suggests that higher temperatures during heat treatment led to darker coloration of the wood. Similarly, the weight of the wood decreased with increasing temperature during the air heat treatment. This reduction in weight could be attributed to the loss of moisture and volatile components from the wood because of the heat treatment process. The relative crystallinity of the wood increased with increasing temperature during air heat treatment. This suggests that higher temperatures led to a greater alignment and ordering of the wood’s molecular structure, resulting in increased crystallinity. These findings suggest that air heat treatment at different temperatures can alter various physical and structural properties of *Paulownia tomentosa* wood [88].

Tang et al. (2019) heat-treated bamboo in tung oil at 100–200 °C, resulting in insignificant color change in vascular bundles in the transverse section but significant darkening as the time proceeded [89].

Lee et al. (2020) tested rubberwood and oil palm trunk particleboards. Samples of 50 mm × 50 mm × 12 mm were firstly put in a palm oil bath and then heat-treated at 180 °C, 200 °C, and 220 °C for 2 h. Color was measured and the results showed that the higher the process temperature, the darker the sample. This darkening was caused by the quinones, the extractives, the amino acids, and the low molecular sugars that migrated to the surface due to heat treatment. Nevertheless, oil also played a significant role because of the layer it formed on the surface [90].

*Paulownia tomentosa* and *Pinus koraiensis* were heat-treated either in hot oil or in hot air and color changing was measured by Suri et al. (2021). The process temperatures and durations were 180 °C, 200 °C, and 220 °C for 1 h, 2 h, and 3 h. The samples treated in oil were darker than those treated in air, especially in *Paulownia tomentosa*. The total ΔΕ increased as the temperature increased for both treatments. However, the treatment duration was reduced in oil heat treatment, making this method more effective in color improvement [91].

*Populus deltoides* were treated with SOHT (silicone oil heat treatment) and ROHT (rapeseed oil heat treatment) as described before. Both SOHT and ROHT treatments resulted in changes in the color of the wood samples. The SOHT-treated samples appeared darker compared to those treated with ROHT. This darkening effect is attributed to various factors such as the formation of colored products from the thermal degradation of hemicelluloses, the emergence of new extractive compounds from wood polymers, and the oxidation of lignin at high temperatures. ΔΕ at ROHT were 42.5 and for SOHT were 51 [65].

### 3.3. Biological Durability

Another important benefit of oil heat treatment is that it improves the resistance against biotic enemies like fungi and termites. This is a result of the hydrophobic properties of oil and the moisture content for fungal growth being delayed, taking into account that moisture is the main factor of fungal growth in wood [86].

Sailer et al. (2000) concluded with similar results, showing also a better resistance against *Coniophora puteana* [47].

In their study, Spear et al. (2006) studied Corsican pine (*Pinus nigra* var. *maritima*) and Norway spruce (*Picea abies*) with sample dimensions of 15 mm × 25 mm × 50 mm. The oils used in this study for OHT were linseed oil, rapeseed oil, and a modified linseed oil resin UZA, at various temperature parameters such as 180 °C and 200 °C and for three hours. Pine at 200 °C showed better results than the equivalent treatments of spruce, giving large differences in treatment WPG (weight percent gain) and fungal decay weight loss between the two species [20].

Lyona et al. (2007), heat-treated Japanese cedar (*Cryptomeria japonica* D. Don) and beech (*Fagus crenata* Blume) in boric acid and rapeseed oil, soybean oil, and linseed oil. Dimensions samples were 20 × 10 × 10 mm for the termite test and 40 × 20 × 5 mm for the fungi test and were immersed in a hot oil bath at 130 °C and a cold oil bath at 80 °C for 1 h each. Decay resistance against *P. sanguineus* was significantly improved and the extent of the improvement was relevant to the temperature and time of the process. Concerning termite resistance, there has been limited research showing controversial results (Lee et al., 2020). However, in Lyona et al. (2007) the results showed that the weight loss after termite attack was lower than in the untreated samples [92].

Fungal decay caused by brown-, white-, and soft-rot fungi is indeed one of the most significant and widespread types of wood degradation. Brown-rot and white-rot are the two major types of wood decay caused by Basidiomycetes, a group of fungi known for their ability to break down lignocellulosic materials [93].

Sidorova (2008) tested spruce, pine heartwood, pine sapwood, and aspen in rapeseed oil at 180 °C, 210 °C, and 240 °C for 0.5 h, 1.0 h, and 1.5 h [94]. Her results are similar to the results of [72] as well as in [70], both mentioned above. In the latter, the palm oil heat treatment on bamboo samples was found to enhance the resistance against fungi and insects.

Bazyar (2012), mentioned above, also studied his samples for decay resistance and showed the same results as [95]. Tjeerdsma et al. (2005) also showed that resistance against brown-rot and soft-rot were improved thusly [96]. On the opposite side, in Baar et al., (2021), thermal treatment does not significantly protect beech against biotic agents, whereas other treatments do significantly protect it [8].

Lacić et al. (2014) tested the biological durability of alder wood (*Alnus glutinosa* L.) after soybean oil heat treatment. The dimensions of the 48 specimens were 15 × 5 × 30 mm and they were heated in an oil bath for 6 and 10 h at 180 °C and 200 °C, respectively. The experiment confirmed that biological durability increased in the modified wood compared to the control specimens. The increase in durability is relevant to the temperature increase and mass loss of specimens. However, there was no significant result concerning the modification time [95].

In the research of Hao et al., 2021, oil heat treatment (140 °C, 160 °C, 180 °C, and 200 C for 2 h, 4 h, and 6 h) of moso bamboo 5 years old with dimensions 300 mm  ×  30 mm  ×  8 mm significantly enhanced the decay resistance and anti-mildew properties of bamboo. Increasing the heat treatment temperature and duration led to further improvements in these properties. The reduction in starch and hemicellulose content, along with changes in surface properties induced by oil heat treatment, contribute to the improved anti-mildew properties of bamboo. These findings provide valuable insights for the development of eco-friendly methods to enhance the durability and resistance of bamboo products against mold and fungal decay [64].

Piao et al., 2022, describe a green and eco-friendly method for modifying bamboo using wood wax oil combined with thermal treatment to enhance its properties. They use 5-year-old moso bamboo (Phyllostachys edulis) with dimensions of 20 mm × 6 mm × 6 mm (longitudinal × radial × tangential). The samples were immersed in wood wax oil at 25 °C and 180 °C for 4 h each. The combination of wood wax oil impregnation and thermal treatment effectively improves the properties of bamboo, making it more suitable for various applications in building materials where enhanced water resistance, fungal resistance, dimensional stability, and mechanical strength are required. The antifungal activity of the untreated and oil-heat-treated bamboo samples was tested in simulated outdoor environments. After 20 days, the bamboo treated with wood wax oil (WWO) at 180 °C showed almost no mold growth on the surface, indicating enhanced antifungal properties compared to the untreated and lower-temperature WWO-treated samples. The infested area of the raw bamboo and WWO-treated bamboo at 25 °C reached 93% and 56%, respectively, while the bamboo treated with WWO at 180 °C showed minimal fungal growth [97].

The research of Suri et al.(2023) highlights the potential of heat treatment, both AHT and OHT, in improving wood durability against fungal decay. Wood samples were subjected to heat treatment using palm oil and air at temperatures of 180 °C, 200 °C, and 220 °C for a duration of 2 h. Both air-heat-treated (AHT) and oil-heat-treated (OHT) wood at 220 °C significantly enhanced the durability of *P. tomentosa* and *P. koraiensis* wood against brown- and white-rot fungi. Significant differences in weight loss between the oil- and air-heat-treated samples were observed, particularly at 220 °C. The heat-treated wood exposed to white-rot fungus exhibited lower weight loss compared to those exposed to brown-rot fungus. The examination of the untreated and heat-treated wood samples revealed that cell components in both *Paulownia tomentosa* and *Pinus koraiensis* at 180 °C were severely damaged after fungal exposure, whereas less damage was observed at 220 °C. The heat-treated wood at 180 °C showed an effect on relative crystallinity following fungal exposure, while this effect was not observed at 220 °C. This indicates that higher temperature heat treatment may better preserve the crystalline structure of the wood. However, further investigation is needed to understand the comparative efficacy of AHT and OHT in fungal resistance [98].

### 3.4. Weathering Durability

The exterior use of wood may cause degradation, and in order to minimize this weathering effect, there are three main practices that could be followed. Firstly, materials with higher durability should be chosen, conforming to EN 350-2 (1994). Secondly, it is required to follow specific measures concerning the shape, the covering of the edges, ventilation, and others [99]. Thirdly, there are several wood surface treatments, like oil heat treatment, that can enhance weathering durability [100]. Degradation can appear, like color changing, crack formations, and increased roughness. The main factors that influence the degradation rate are water and UV radiation [101].

Weathering tests last for a significant amount of time, either measured in hours or months, depending if it is a natural process or not. Unfortunately, there is little research on the weathering durability of hot oil-treated wood.

Even though oil-heat-treated wood can be glued or varnished without any problem, it is not highly appropriate for exterior use because it is considered of low weathering resistance [102].

Bak et al. (2012) investigated *Populus euramericana cv. Pannónia* and *Robinia pseudoacacia* L. wood samples that were heat-treated in sunflower, linseed, and rapeseed oil at 160 °C and 200 °C for 2 h, 4 h, and 6 h. During weathering, the color was measured directly after the treatment and every 30 days for 12 months. The weathering procedure is shown in the next figure. The samples were put on wooden frames, facing south, at 45°. There was less color change in the treated samples. However, for better protection in the long term, it is recommended to use surface finishing [71].

Nejad, Dadbin, and Cooper (2019) studied coating performance on oil-heat-treated spruce and southern yellow pine wood samples. Color was measured directly after the treatment and after 30 days for the first 3 months and then every 6 months for a total of 18 months of natural weathering. There was a lower color change and a better appearance in the treated samples [103].

Jamali and Evans (2020) studied the performance of an acrylic coating on 240 mm × 38 mm × 89 mm samples of blue-stained lodgepole pine wood after heat treatment in an oil bath at 220 °C for 2 h and plasma treatment for 33 s. The natural weathering process lasted for 18 months and resulted in darker-colored wood, especially in the hot oil-treated specimens [104].

Suri et al. (2023) investigated the effects of artificial weathering on the surface features, physical qualities, and chemical compositions of oil-heat-treated (OHT) and air-heat-treated (AHT) *Paulownia tomentosa* and *Pinus koraiensis* woods. The experiment treated untreated and heat-treated samples at 180 °C, 200 °C, and 220 °C for 2 h with ultraviolet (UV) and water for 168 h and 336 h, respectively, according to ASTM G53-96 standards. The weathering experiment consisted of 2 h of UV-light irradiation followed by 2 h of condensation cycles. The heat-treated woods, both OHT and AHT, showed much lower overall color change and volumetric swelling than the untreated woods [105].

Both the untreated and AHT woods at 180 °C exhibited a golden-yellow color after weathering. The AHT *P. tomentosa* woods at 200 °C and 220 °C showed a grayish color, while the AHT *P. koraiensis* woods at the same temperatures showed a dark-yellow color. The AHT woods at 200 °C and 220 °C showed uneven color distribution after weathering. In contrast, the OHT woods showed negligible color differences before and after weathering, with a more uniform color distribution, likely due to the even surface protection provided by oil heat treatment. The OHT woods generally showed significant increases in VS after weathering, indicating dimensional changes. The OHT woods demonstrated better dimensional stability compared to the AHT woods, with the highest stability observed in the OHT wood treated at 220 °C. The study underscores the effectiveness of oil heat treatment in maintaining color stability and preserving the appearance of wood species subjected to artificial weathering, with implications for enhancing the performance and longevity of wood products in outdoor applications.

## 4. Mechanical Properties

The treating medium is a critical factor in the mechanical properties of heat-treated wood. Air, for example, has a lesser effect on bending strength than oil. After oil heat treatment, elasticity seems to be increased and brittleness decreased [6].

Several studies mentioned in previous chapters also assessed the mechanical properties of wood after oil heat treatment. Tjeerdsma et al. (2005) showed that oil heat treatment affected the elasto-mechanical properties and that the type of oil played a significant role in the extent of the changes [96]. Hao et al. (2021) study resulted in the increased compressive strength and bending strength of bamboo samples [64]. In Mohebby et al. (2014), the mechanical strengths, apart from impact load resistance, are improved, in contrast to Wahab et al. (2017) and Baar et al. (2021), who showed reduced strength, even though oil heat treatment resulted in stiffer beech wood in bending in the latter’s research [60,77]. Lee et al. (2020) concluded that mechanical strength is also reduced in particleboards after heat treatment and this is due to the low heat resistance of the UF resin [90]. In Bak and Nemeth (2012), the brittleness of the samples increased as the impact bending values decreased [71].

Manalo and Acda (2009) tested three species of bamboo, *Bambusa blumeana*, *Bambusa vulgaris*, and *Dendrocalamus asper* in virgin coconut oil at 160–200 °C for 30–120 min. The strength properties, elasticity, rupture, and toughness, were decreased in correlation to the temperature. The duration of heat treatment played no significant role [106].

Avila et al. (2012) and Fang et al. (2012) took samples of rotary-peeled aspen (*Populus tremuloides*) wood veneers of 50 mm × 50 mm × 3.2 mm and the first immersed them in hot canola oil at 200 °C and 220 °C for 2 h and the second at 180, 200 and 220 °C for 1, 2, and 3 h, respectively. There was a significant increase in the mechanical properties as a result of the increase in the wood density. In the second study, there was a decrease in Brinell hardness and tensile strength and a slight increase in bending strength [107,108].

Another research was performed on *Pinus sylvestris* samples which were impregnated with water glass, melamine, silicone, or tall oil and then heat-treated at 180 °C and 212 °C for 3 h (Lahtela and Kärki, 2016). Mechanical strengths were impaired as the temperature increased. A general conclusion of this research was that all properties were enhanced when heat treatment took place after impregnation [109].

Cheng et al. (2014) studied poplar (*Populus* spp.) 22 mm × 160 mm × 260 mm samples after soybean oil heat treatment at 180 °C for 2 h. There was an increase in the compression strength parallel to the grain and this was due to the weight gain after the oil uptake [110].

Taşdelen, Can, and Sivrikaya (2019) investigated *Pinus pinaster marittima* and *Populus euroamericana* samples with heat treatment in safflower, linseed, and hazelnut oil at 160 °C, 180 °C, and 200 °C for 2 h, 4 h, and 6 h, respectively. The compression strength parallel to the fiber direction increased.

*Fagus sylvatica* L. wood was thermally modified in Reinprecht and Repák’s (2019) research in paraffin at 190 and 210 °C for 1 h, 2 h, 3 h, or 4 h. The mechanical properties of the samples were deteriorated. Both impact bending strength and Brinell hardness decreased. The impact bending strength and Brinell hardness of wood samples subjected to paraffin-thermal modification treatments were assessed under various conditions. The reference treatment yielded an impact bending strength of 5.38 J.cm^−2^ and a Brinell hardness of 31.56 MPa. When treated with paraffin only, the impact bending strength decreased to 4.27 J.cm^−2^, while the Brinell hardness remained relatively unchanged at 32.22 MPa. Treatment at 190 °C for 1 h resulted in an impact bending strength of 3.76 J.cm^−2^ and a Brinell hardness of 30.81 MPa, respectively. Increasing the duration to 2 h yielded higher impact bending strength (4.42 J.cm^−2^) and Brinell hardness (29.09 MPa) (Reinprecht and Repák, 2019). Based on the provided data of paraffin on the impact bending strength and Brinell hardness of the materials subjected to paraffin-thermal modification under different conditions, we can draw that the impact bending strength generally decreases with the duration of thermal modification. The samples treated with paraffin alone have lower impact bending strength compared to the reference. The impact bending strength tends to decrease as the temperature of thermal modification increases. Brinell hardness also tends to decrease with the duration of thermal modification. The samples treated with paraffin alone have a similar Brinell hardness compared to the reference. Brinell hardness significantly decreases with the increasing temperature and duration of thermal modification. So, longer durations and higher temperatures of paraffin-thermal modification tend to result in reduced mechanical properties, as evidenced by decreases in both impact bending strength and Brinell hardness [111].

Koumbi-Mounanga et al. (2020) experimented on 10 × 25 × 25 mm American beech (*Fagus grandifolia* Ehrh.) and pin oak (*Quercus palustris*) species. The samples were heat-treated in soybean oil at 220 °C for 2 h and then conditioned without oil at 103 °C. The bending strengths were reduced [46].

The findings reported by Suri et al. (2021:2022:2023) indicate that oil-heat-treated (OHT) specimens exhibit several advantageous properties compared to air-heat-treated specimens. Lower weight loss in abrasion: the OHT specimens demonstrated less weight loss under abrasion compared to the air-heat-treated specimens. This indicates that the oil treatment may have enhanced the wood’s resistance to wear and abrasion, potentially prolonging its lifespan in applications subject to mechanical stress. The density of the OHT specimens was higher than that of the air-heat-treated specimens. This could indicate that the oil treatment led to greater wood compaction or reduced porosity, resulting in higher density. The OHT specimens exhibited higher compressive strength compared to the air-heat-treated specimens. This suggests that the oil treatment may have improved the wood’s ability to withstand compressive forces, which could be advantageous in structural applications. The OHT specimens showed greater hardness compared to air-heat-treated specimens. Increased hardness can enhance the wood’s resistance to indentation and deformation, making it suitable for applications where durability is important [91,98,105,112].

In the research of Piao et al., 2022, the results suggest that wax oil heat treatment, particularly at 180 °C, effectively enhances the mechanical properties of bamboo, making it harder and increasing its load-bearing capacity. When the bamboo was treated with WWO at room temperature, the modulus of elasticity (MOE) of the treated sample was like that of untreated bamboo. However, the modulus of rupture (MOR) increased, indicating an improvement in the load-bearing capacity. When the bamboo was treated with WWO at 180 °C, both the MOE and MOR were maintained at 2036.39 MPa and 83.88 MPa, respectively, and remained higher than those of untreated bamboo [97]. The observed phenomenon can be attributed to the specific chemical interactions that occur within the wood structure at 180 °C. Specifically, at this temperature, the bonding of glucomannan in hemicelluloses to the cellulose surface predominantly relies on hydrogen bonds, while lignin forms covalent bonds with hemicelluloses to a limited extent. As a result of these processes, while the density of the samples remains relatively unchanged, exposure to a wood oil treatment (WWO) at 180 °C results in the degradation of hemicellulose and a reduction in the cohesive bonds between different cell wall components. However, the simultaneous formation of new chemical bonds contributes to enhanced tensile strength. It is notable that the tensile strength of the samples treated with WWO at 180 °C reaches approximately 130 MPa, representing an increase of approximately 30 MPa compared to the untreated or room temperature-treated samples.

Kaya et al., 2023, investigated the effects of linseed oil and heat treatment on the physical, mechanical, and acoustic properties of cypress and maple wood. Heat treatment was conducted at four different temperatures (160 °C, 180 °C, 210 °C, and 240 °C) in an oven. The results indicated significant enhancements in the physical, mechanical, and acoustic properties of the treated wood compared to the control samples. For Mediterranean cypress wood, the maximum decrease in the modulus of rupture (MOR) was observed at 240 °C, with a reduction of 27%. Similarly, the modulus of elasticity (MOE) decreased by 23% at the same temperature. However, the compression strength (CS) increased by 9% at 160 °C, although this increase tended to decrease with higher temperatures, reaching not more than 2% at 210 °C. This suggests that the impact of the heat treatment on Mediterranean cypress wood was more pronounced than that on field maple wood. The decrease in density was observed for both types of wood, with Mediterranean cypress wood experiencing a loss of 18% at 240 °C, while field maple wood lost 30% of its density. This decrease is attributed to the removal of water from the wood structure and degradation, particularly of hemicellulose components. Comparing the impregnated samples with untreated ones, it was observed that as the temperature increased, the mechanical properties such as MOR and MOE increased, although CS did not change significantly. Specifically, for Mediterranean cypress wood, the maximum increase in MOR was 8% at 240 °C, while MOE increased by 4% at the same temperature. For field maple wood, both MOR and MOE increased by 13% at 240 °C, with CS increasing by 7%. These enhancements indicate improved strength and stiffness properties, which are vital for structural applications [81].

## 5. Discussion

A thorough review of the literature indicates a growing interest in utilizing oil heat treatment as an environmentally friendly method for modifying wood. While existing research has significantly contributed to our understanding of this process, there are areas that require further exploration and refinement. One significant observation is the variability in oil heat treatment procedures, encompassing differences in wood species, oil type, temperature, and the duration of the process. This diversity underscores the complex relationship between treatment parameters and resulting wood properties. Thus, future studies should prioritize the standardization of methodology to facilitate clearer comparisons and generalizations across experiments.

Alterations in chemical wood constituents induced by the treatment process, such as a decline in cellulose and hemicelluloses coupled with an increase in lignin content and its network, suggest enhanced thermal stability, dimensional stability, and biological resistance. A loss or alteration of some extractives is implemented during the process, altering the appearance and other extractive-related properties of wood. However, a deeper investigation into the mechanisms driving these changes, particularly the interactions between wood components and oil mediums, is required. Of particular importance are the type and stability of the bonds that are developed between the components of the wood and the molecules of the protective medium/substance, the resistance of the modified wood to leaching effects under specific temperature and relative humidity conditions, and the properties that the wood acquires after the application of the modification. From a physical standpoint, improvements in cellulose crystallinity, the depolymerization of amorphous areas of cellulose, as well as of hemicellulose areas, in combination with the dimensional stability post-treatment offer promising prospects for utilization in wood engineering applications.

Nevertheless, understanding the correlation between treatment parameters and these physical enhancement changes is crucial for optimizing the process across diverse wood species and end-use scenarios. While the existing literature suggests enhanced resistance to biotic degradation in oil-heat-treated wood, there is a noticeable gap in research on weathering durability, performance, and the duration of treated wood in time. Addressing this shortcoming is essential for a comprehensive evaluation of the long-term performance of oil-heat-treated wood in outdoor environments.

Conflicting findings regarding strength enhancements and reduction in brittleness highlight the complexity of the interaction between oil impregnation and wood mechanics. Future investigations should delve into the underlying mechanisms to reconcile these discrepancies and optimize treatment protocols for desired mechanical properties. Comparative, detailed, in-depth studies based on the same wood species and providing comparable results on the most beneficial and promising methods of thermally modifying wood employing oils are necessary to get even closer to drawing safe conclusions about the properties of modified wood, and the utilization potential and the possible range of future applications.

## 6. Conclusions

In conclusion, the reviewed literature underscores the effectiveness of oil heat treatment as a sustainable method for enhancing wood properties. The presence of specialized companies and the availability of non-toxic, cost-effective oil mediums further highlight the practical feasibility of this approach. Various processes and their effects on the chemical, physical, mechanical, and biological properties of wood have been examined.

Vegetable oils such as linseed, soybean, and palm, known for their non-toxic and cost-effective nature, present optimal choices as heat transfer media in the oil heat treatment process, given their higher boiling points compared to treatment temperatures. However, the efficacy of oil heat treatment processes hinges on factors such as wood species, sample dimensions, oil type, heat temperature, and process duration.

Regarding chemical properties, most of the literature indicates a gradual decrease in cellulose and hemicellulose content, alongside an increase in lignin content, which enhances thermal stability. Physically, increased cellulose crystallinity contributes to reduced equilibrium moisture content (EMC) and improved dimensional stability. Additionally, higher treatment temperatures result in darker wood color, influenced by the type of oil used.

Overall, oil heat treatment improves wood’s resistance against biotic threats such as fungi and termites, owing to the hydrophobic properties of oil and delayed moisture content for fungal growth. Nonetheless, further research is warranted to assess the weathering durability of oil-heat-treated wood comprehensively.

While most of the literature indicates an increase in mechanical strength and a decrease in brittleness compared to heat treatment without oil impregnation, standardizing methodologies and addressing knowledge gaps, particularly concerning weathering durability and mechanical behavior, are imperative for fully leveraging the potential of oil heat treatment. By doing so, the oil heat treatment of wood can evolve into a cornerstone for producing greener, high-value wood products that meet industry and environmental standards.

## Figures and Tables

**Table 1 materials-17-02394-t001:** Wood modification—different processes [12].

Wood Modification										
Chemical Processes		Hydro Thermo Mechanical Processes			EMR- and Plasma-Based Processes			Other Processes		
Active Treatments	Passive Treatments	Thermo Hydro Treatment	Thermo Mechanical Treatment	Thermo-hydromechanical Treatment	Electromagnetic Radiation	Laser Treatment	Plasma Treatment	Biological-Based Processes	Metabolite Treatment	Bio mimicry Processes
Acetylation, Furfurylation other types of chemicals, and DMDHEIJ-based processes	Impregnation type processes melamine, natural oils, and Polyethylene glycol	Releasing internal stresses	Self-bonding of veneer wood welding	Shaping, Molding, and Densification	Ultraviolet light, infrared light, and high frequency Microwaves	CO_2_ laser and UV laser ablation	Cold plasma and hot plasma	Microbial treatment with Aerobic spore-forming bacteria. Fungi antagonists	Microbial metabolite and Enzymatic activation	Salt-based processes

**Table 2 materials-17-02394-t002:** Chemical modification processes typically used for wood species, process parameters (Temperature and pressure), and properties (http://www.diva-portal.se/smash/get/diva2:1525440/FULLTEXT01.pdf accessed on 13 May 2024).

Species Used	Process	Process Temperature (°C)	Pressure (Mpa)	Properties	
Radiata pine	Accoya^®^	170	2.2	Improved dimensional stability, durability, low mold resistance, and corrosive to fastener	[17]
Norway spruce, Scots pine, and various others	Thermal modification (various)	160–230		Improved dimensional stability, moderate durability, and reduced mechanical properties	[18]
European beech and Scots pine	Impreg™ (MF) (various)	60–150	0.005–1.2	Improved durability, less swelling/shrinking, and lower hardness and toughness	[19]
Norway spruce and Scots pine	Organowood^®^	20–120	1.6	Stable silver-grey surface and improved durability	[20]
Radiata pine and Scots pine	Kebony^®^ and Nobelwood^®^	140	1.2	Improved durability, dimensional stability, hardness, corrosion to fasteners, and lower toughness	[12]
European beech and Scots pine	Chitosan	20–80	0.01–1.2	Improved durability	[21]
Oak	PEG	20–60	0.1	Lower moisture uptake and swelling, and lower stability	[22]
Scots pine and Norway spruce	Linotech™	60–140	0.8–1.4	Improved durability and oil exudation	[23]

**Table 3 materials-17-02394-t003:** Examples of thermal modification processes and their curing conditions (Sandberg et al., 2017) [25].

Process	App. Year	Temperature (°C)	Duration (h)	Inert Atmosphere	Ref.
FWD	1970	120–180	15	steam	[41]
Plato	1980	150–180/170–190	4–5/70–120	steam/heated air	[42]
ThermoWood	1990	130/185–215/80–90	30–70	steam	[43]
Le Bois Perdure	1990	200–230	12–36	steam	[42]
Rectification	1997	160–240	8–24	nitrogen or other gas	[42]
OHT	2000	180–220	24–36	vegetable oils	[42]
ThermoVuoto (TVT)	2010	160–220	<25	vacuum	[44]

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
